# MFC/NFC-Based Foam/Aerogel for Production of Porous Materials: Preparation, Properties and Applications

**DOI:** 10.3390/ma13235568

**Published:** 2020-12-07

**Authors:** Chenni Qin, Mingzhu Yao, Yang Liu, Yujie Yang, Yifeng Zong, Hui Zhao

**Affiliations:** 1College of Light Industry and Food Engineering, Guangxi University, Nanning 530004, China; Qinchenni@st.gxu.edu.cn (C.Q.); yaomingzhu@st.gxu.edu.cn (M.Y.); 2016301040@st.gxu.edu.cn (Y.Y.); 2016391055@st.gxu.edu.cn (Y.Z.); zhh@gxu.edu.cn (H.Z.); 2Guangxi Key Laboratory of Clean Pulp & Papermaking and Pollution Control, Guangxi University, Nanning 530004, China

**Keywords:** cellulose, foam, aerogel, Pickering foam, capillary foam

## Abstract

Nanofibrillated cellulose and microfibrillated cellulose are potential raw materials separated from plant fibers with a high aspect ratio and excellent mechanical properties, which can be applied in various fields (packaging, medicine, etc.). They have unique advantages in the preparation of aerogels and foams, and have attracted widespread attention in recent years. Cellulose-based porous materials have good biodegradability and biocompatibility, while high porosity and high specific surface area endow them with strong mechanical properties and liquid retention performance, which can be used in wall construction, sewage treatment and other fields. At present, the preparation method of this material has been widely reported, however, due to various process problems, the actual production has not been realized. In this paper, we summarize the existing technical problems and main solutions; in the meantime, two stable systems and several drying processes are described, and the application potential of cellulose-based porous materials in the future is described, which provides a reference for subsequent research.

## 1. Introduction

At present, cellulose-based multi-porous foams are mainly divided into cellulose-based aerogels and cellulose-based foams. In some literature, the two are also used interchangeably [1]. In the previous report, aerogel was defined as “a highly porous solid of ultra-low density and with nanometric pore sizes formed by replacement of liquid in a gel with gas” [2]. Foam is defined as “solid porous materials with microscopic pores” [3]. Aerogels are generally described as porous materials with high porosity and made by freeze drying or supercritical drying using nanofibrillated cellulose as the substrate [4,5,6]. While cellulose foam is used to describe multi-porous materials based on MFC (microfibrillated cellulose) or NFC (nanofibrillated cellulose) with high porosity, which is made by oven- or freeze drying, however, the pore size is larger [7,8,9]. In this paper, we call it “foam”. Previously, the research on cellulose mainly focused on the field of composite materials. In recent years, as people have gained a deeper understanding of NFC and MFC, it has been discovered that due to their nanoscale size and high aspect ratio, both of them can be used as a stable structure in the foam so as to form “Pickering foam” [8,10]. Different from the traditional Pickering foam, the hydrophobic or hydrophilic nanosolid particles are replaced by MFC and NFC, and fibers will wrap bubbles. Due to the electrostatic repulsion of fibers, the thickness of bubble liquid film will not be thinned rapidly over a period of time and maintains a certain scale, thus slowing down the foam coalescence. At the same time, fibers will gather in plateau channel, which hinders the diffusion of gas in bubbles and slows down the coarsening of bubbles [11]. MFC is a mixture of cellulose with different morphology and size, which is mainly obtained from plant fiber by mechanical treatment. It is a basic fibril bundle composed of less-ordered regions. Its main component is NFC (within 100 nm in diameter), and there are also fiber fragments and microfilaments with diameters ranging from several hundred nanometers to several microns. Due to the thixotropic viscosity characteristic of NFC, it will produce “gelation” reaching a certain concentration to increase the viscosity of the system and slow down the “Ostwald ripening” of foam.

Maintaining the network structure of foam or minimizing its degree of damage is a common problem in the process of preparing light and strong solid porous materials. According to a large number of literature reports, there are two main ways to solve this problem, one is to increase the stability of wet foam, and the other is to obtain solid foam with 3D network structure by optimizing the drying process. The former is the premise of the latter. For wet foam, to enhance the stability of foam is to establish Pickering foam and capillary foam by adding surfactants, polymers or nanoparticles to bulk solution. Drying methods are mainly divided into freeze drying [12], supercritical drying [13] and oven drying [7,8,10]. Freeze drying is most commonly used, but because of its complex operation and high cost, it is not suitable for large-scale production. At present, it remains in the laboratory stage. Supercritical drying is not widely used because of its complex operation and high requirements on equipment. Compared with the former two methods, the natural air drying or oven drying, the operation is simple, and it is easy to realize industrial production, but the pore size will be larger, and the network structure of foam will be easily damaged due to the existence of gas–liquid interface pressure during drying. At present, cellulose-based foam is still staying in the laboratory, and has not yet achieved industrial production. Compared with EVA (ethylene vinyl acetate) and EPS (expanded polystyrene) foam boards which already exist and are widely used in the market at present, the porous foam materials made of cellulose are biodegradable, non-toxic and harmless, and environmentally friendly. Different preparation methods of cellulose-based porous foams with different properties have been widely reported in the literature up to now, and the potential applications in various fields have been expounded. For example, it can be used as a cushion for portable devices due to its light weight, firmness and porosity. At the same time, it can also be applied to biomedical applications such as wound dressing [14], tissue scaffolds [15,16,17,18,19] and drug release [20,21,22,23,24], as shown in Figure 1. In addition, the foam porous structure can be used for liquid absorption [25,26,27] such as sewage treatment and water–oil separation [28,29,30,31]. Some porous materials also have low thermal conductivity, which can be used as thermal insulation materials [2,9,32,33,34,35,36]. Also, foam can be used as sound insulation [37,38] and materials to make wall sandwich panels. However, at present, cellulose-based foam materials are still in the research stage and have not been put into industrial production.

Figure 2 summarizes the published quantity of cellulose foam materials in the past 10 years, and the data are provided by the Science Direct database. It can be seen that more and more attention has been paid to cellulose-based porous materials, and there is a great space for development. The foam forming is a novel technique for producing bulk materials with low density and high porosity. It is a comprehensive, well-structured approach. As regards the other review articles already published on this topic, this paper describes two stabilization mechanisms of MFC/NFC stabilized wet foam: Pickering foam and capillary foam, in which MFC and NFC play the role of stabilizing particles, At the same time, we introduce the main mechanisms of both, the recent research progress and potential application prospects of cellulose porous foams in recent years are reviewed, and the main problems and solutions for the preparation of cellulose porous foams are also summarized. In particular, we describe two stable systems for wet foam: Pickering foam and capillary foam., so as to provide a reference for future research. 

## 2. Preparation of Porous Foam Materials

### 2.1. Process from Suspension to Solid Foam

The preparation process from suspension to solid foam is generally divided into five steps (as shown in Figure 3). The first step is the preparation of fiber suspensions. Cervin et al. [7] used modified NFC as a stable particle, and in order to increase the adsorption of fibers at the interface, octylamine was added to the suspension, and the wet foams were formed by high-speed stirring, then the foams were placed on the pre-soaked porous ceramic sheets, and finally dried in an oven to obtain the solid foams.

### 2.2. Preparation of Nanofibrillated Cellulose (NFC) and Microfibrillated Cellulose (MFC)

Over the years, cellulose has been involved in a wide range of fields, especially NFC and MFC, which have the characteristics of ultra-light, high porosity and high strength, and have been used in composite materials in the past [39]. In recent years, with the shortage of petroleum resources and the improvement of people’s awareness of environmental protection, cellulose-based foam materials have become potential materials to replace plastic foam boards in the market. Cellulose is abundant in algae, invertebrates and some bacteria [40], and also exists in the primary wall in the cell walls of plants such as hemp, wheat straw, rice straw and bagasse. Moreover, the cell walls of plants also contain lignin, hemicellulose and other accompanying materials, which wrap cellulose. When extracting NFC and MFC, it is necessary to remove these associated substances to expose cellulose, so pretreatment is usually added. NFC and MFC have high aspect ratio and excellent mechanical properties which can be degraded and regenerated [39]. It is filamentous in morphology, but in practical application, it is a network structure formed by winding, interweaving or connecting nanometer or micron filaments. When dispersed in polar liquids, it can expand into a smooth gel with thixotropic viscosity properties in the suspension. It is a stable gel that can be used for storage or to withstand freeze-thaw cycles.

There are many preparation methods for NFC and MFC. Figure 4 shows the steps of extracting MFC and NFC from natural plants. The preparation methods of both are similar, and the main difference is related to the size of the extracted cellulose nanofibers. The diameter of MFC is generally in the micron level, and the length is mainly distributed in the micron or even mm, while the relative diameter of NFC is mainly distributed in the nanometer level, and the length distribution mainly depends on the subsequent craft. NFC usually adopts high-pressure homogenization, which usually requires high-pressure homogenization to make its particle size distribution small and uniform, while mechanical grinding is the most common and easy to extract MFC. Their preparation methods can be roughly divided into chemical methods, biological methods and mechanical methods. In the actual extraction process, pretreatment methods [41,42] (acid, alkali, enzyme and oxidation, etc.) are combined with the above three methods to improve the ratio of length to diameter and the yield of fibers, and at the same time reduce the energy consumption of the whole process. Table 1 compares MFC and NFC made by different raw materials and methods.

It can be seen from Table 1 that enzyme treatment and TEMPO-mediated oxidation (2,2,6,6-Tetramethylpiperidinooxy) are the most commonly used pretreatment methods in practical application. Martoïa et al. [3] used birch bleached kraft pulp as raw material to obtain fibers with different size distributions through enzyme treatment and high-pressure homogenization, in which the smallest nanofiber diameter was 20–50 nm, while the larger microfiber diameter reached 100–500 nm. Yang et al. [43] obtained nanofibers with a diameter of 3 nm and a length of 1.2 um through TEMPO oxidation combined with mechanical treatment, with an aspect ratio of 400. Compared with enzyme treatment, the size distribution of cellulose obtained by TEMPO oxidation is more uniform, and the diameter is basically controlled within 100 nm, while the size distribution of fiber obtained by enzyme treatment is wider, and both nanofibers and microfibers exist. In addition, some people put forward a new pretreatment method. In 2014, it was proposed that catalyst pre-loaded Fenton oxidation method combined with high pressure homogenization method was used to prepare MFC from hardwood dissolving pulp [48]. The concept of “micro-reactor” was put forward, which made up for the shortcoming that H_2_O_2_ in a traditional Fenton oxidation system had invalid decomposition before effectively oxidizing fibers, and improved the production efficiency. The yield was also high, and the size distribution of the prepared MFC was relatively uniform, with a diameter of about 200 nm and a length of tens or even hundreds. In addition, some people added chemical reagents to optimize the whole extraction process based on the above methods. In 2015, Yue et al. [50] used bleached coniferous wood chemical pulp as raw material to prepare NFC by synchronous method. This method used ZnCl_2_ with swelling effect as pretreatment agent, and realized swelling and mechanical dissociation simultaneously when swelling was combined with moderate mechanical pulping treatment.

Among many treatment methods, the mechanical method (ultrasonic crushing, high-pressure homogenization and grinding method) is the most commonly used, which does not need to use chemical reagents and is widely used for environmental protection. However, due to its high energy consumption, cellulose is usually pretreated (enzyme pretreatment, TEMPO oxidation) before this to degrade it partially, so as to reduce energy consumption. The chemical method uses acid or alkali to decompose cellulose into nanoscale or micron cellulose, which occurs in the amorphous region of cellulose, because the structure of this area is loose and easy to degrade. However, this method is not widely used because of the large amount of waste liquid, low yield and high cost. Biological methods are mainly enzyme treatment methods, which hydrolyze cellulose under mild conditions for a long time. In fact, this method is usually combined with other methods to improve production efficiency. In addition, biological methods include microbial synthesis of bacterial cellulose (BC), which has the advantages of excellent three-dimensional network structure and high hydrophobicity, and also has unique physical and mechanical properties [51].

### 2.3. Preparation of Wet-Foam

#### 2.3.1. The Processing from MFC/NFC Dispersions to Wet Foam

Mechanical stirring (the so called Bartsch method) [8] is the most commonly used method to introduce bubbles into solutions (surfactants are usually added). Meanwhile, in the Bikerman method [11,52] gas (such as nitrogen, Carbon dioxide) can be introduced to foam by a pressure device. In addition, foam can be generated by adding chemical reagents (such as 4-methylbenzenesulfonhydrazide and Diethyl azodicarboxylate) [53,54], which are decomposed after heating. However, this method may cause environmental pollution. Among the above three methods, mechanical stirring is widely used because of its simple operation and no pollution in the environment. For mechanical foaming, the choice of foaming agents, stirring rate and foaming agent content (surfactant concentration slightly higher than the critical micelle concentration) are very important. Foaming agents are roughly divided into two types. One is surfactants (the structure of different surfactants and their respective surface tensions give them different foaming properties). Surfactants, because of their strong foaming ability, are most widely used, while proteins are only mainly used in food [55]. Although its foaming ability is mediocre, its stability is better than that of the surfactants. Secondly, during the foaming process, the stirring rate is also very important, the faster the rotating speed, the shorter the foam time to reach the maximum volume. When the rotation speed is low, the air is entrained into the suspension. Large bubbles will form, are gradually sheared and decomposed into smaller bubbles, finally shear thinning occurs. If a pressure device is used to force a gas into a liquid to produce a bubble, the most important of which is the selection of the gas. During the formation of the bubbles, the foaming agent molecules are closely aligned at the gas-liquid interface and envelop the gas. The decay of the bubbles is largely due to the coalescence and coarsening of the bubbles (more detail is described below), both of which are derived from the diffusion and escape of the gases in the bubbles. Therefore, the use of less water-soluble gases gives better stability, and bubbles can remain spherical for longer periods of time.

#### 2.3.2. Wet Foam Stabilization Mechanism

Foam is a metastable system, which is prone to decay under certain conditions, that is, foam instability. There are three causes of foam instability, among which liquid drainage mainly occurs in Plateau channel, under the action of gravity and surface tension, the liquid in bubbles will tend to be discharged to the outside world; Bubble coalescence is mainly due to the weakening of the liquid film due to the drainage intensity, the bubble becomes unstable and finally coalesces; According to the Laplace equation, bubble coarsening is because the gas pressure in small bubbles is greater than that in large bubbles, and there is a pressure difference between them. Gas diffusion occurs in small bubbles and enters large bubbles. Eventually, small bubbles get smaller and disappear, and large bubbles get bigger and eventually rupture. The basis of evaluating foam stability mainly includes liquid film thickness, bubble size, and foam drainage rate [11,56]. With the extension of time, bubble volume will gradually increase, drainage will be accelerated, liquid film will become thinner and thinner, bubble coalescence will occur, and finally bubbles will rupture. The structure of foam is complex, and there are many influencing factors, such as the viscosity of the solution, viscoelasticity of the gas–liquid interface, and selection of the stabilizer.

Based on the instability mechanism mentioned above, there have been a large number of reports on how to improve foam stability. The main methods can be summarized into three types: one is to stabilize by surfactants [8,11,57], the other is to stabilize by polymers [58,59], and the other is to stabilize by hydrophobic or hydrophilic nanoparticles [60,61]. Surfactants are mainly due to having bipolar groups (hydrophilic group and hydrophobic group). In the foam forming process, surfactants will be arranged in an orderly way on the surface of liquid film wrapping gas, in which hydrophilic group points to water and hydrophobic group points to air [62]. When the concentration is low, it mainly exists in the form of molecules at the interface. When the critical micelle concentration is reached, the interface adsorption reaches saturation and micelles are formed. Its stability mechanism is mainly through electrostatic repulsion, which is characterized by strong foaming ability, but bubbles are less stable and prone to collapse. When polymers are added as stabilizers, some polymers will interact with surfactants [57,63]. When the concentration of polymers in surfactants reaches the critical aggregation concentration (CAC), aggregates will be generated and stay at the channels and nodes, which slows down the drainage of bubbles and the growth of bubbles. At the same time, due to the viscosity of polymers, “particle bridging” will occur, which will wrap hydrophilic or hydrophobic nanoparticles on the surface of liquid membrane and connect with each other, thus promoting the formation of networks and increasing the viscosity of the gas–liquid interface.

The Figure 5 shows the stabilization mechanisms of wet foam at present, which are mainly divided into two types: one is the Pickering foam system (Figure 5a), and the other is the Capillary foam system (Figure 5b). In these two systems, anionic surfactant is used as foaming agent (usually, the number of carbon atoms is 12 or 14. When the carbon chain is short, the formed surface film has low strength, and when the carbon chain is long, it will be difficult to dissolve, and the formed surface film is too rigid to maintain stability), among which sodium dodecyl sulfate (SDS) is the most common one, while NFC or MFC are used as stable particles.

#### 2.3.3. Pickering Foam of MFC and NFC

In recent years, with the development of nanotechnology, the use of nanoparticles to stabilize foam has become the mainstream. Compared with the porous materials made by ionic liquid and chemical crosslinking, the foam made by Pickering foam has higher foaming rate and is more stable because of the stabilization of particles. Pickering foam is a kind of wet foam stabilized by small solid particles, which are adsorbed on the interface between two immiscible phases (water/oil or air/water) [64], Gonzenbach et al. [65] used this method to prepare super-stable foam, while cellulose based Pickering foam uses NFC and MFC as solid particles to be adsorbed on the interface of two phases to achieve stability. Figure 5a shows Pickering foam with MFC and NFC as stabilizing particles, which are enriched in the outer side of bubble liquid film and plateau channel, and wrapped around the bubble to form a protective granular layer [66], which reduces the interference of external fluid to the inner environment of bubble. The stability mechanism of the system mainly comes from changing the rheology of bulk solution and gas-liquid interface. MFC and NFC have thixotropic viscosity characteristics, which will form stable gel after absorbing water and swelling, which makes the fibers more tightly adsorbed and entangled on the surface of the liquid membrane to form a stable network structure, enhances the interfacial viscoelasticity, prevents the deformation (thinning) and rupture of the liquid membrane, and slows down the drainage of the foam. Secondly, due to the increase of fluid volume and the increase of viscosity of the bulk solution, the free movement of active molecules is reduced, resulting in the decrease of desorption rate of surfactant at the interface, and the bubble is in a relatively stable state within a certain period of time. Xiang et al. obtained the corresponding storage modulus (G’), dissipation modulus (G”) and complex viscosity (η*) by comparing the viscoelasticity of gas-liquid interface under the oscillation of only SDS and the coexistence of SDS and NFC [11,57]. The results show that NFC does play a thickening role and endows the gas-liquid interface and bulk solution with certain viscoelasticity, confirmed the above statement. In 2013, Cervin et al. [7] developed Pickering foam using NFC as a stabilizer to produce cellulose foams with good mechanical properties, and proposed that the presence of particles resists the shrinkage and growth of the bubble volume compared to a surfactant-based system, Pickering foam greatly increases the foam life cycle and exhibits excellent stability. The final foam density is 0.03 g cm^−3^, the porosity is up to 98%, and the pore size is about 500 um. At the same time, the mechanical properties of the foam were greatly enhanced, and its compressive strength was as high as 437 ± 63 kPa. Using MFC/NFC as the particle stabilizer, Liu et al. [67] prepared a three-dimensional ultra-lightweight foam with a density of 0.1 kg cm^−3^ and a porosity of 90%. It is found that Pickering foam can greatly increase the lifetime of foam [66], which is beneficial to the application of foam-cleaning products. However, it is far from enough for the preparation of porous solid foam with high porosity and cross-linked network structure to maintain its existing network structure when dried. Therefore, on this basis, Koos put forward the concept of capillary foam in 2014, and found that the addition of secondary liquid can significantly improve the stability of foam and reduce the collapse of structures [68].

#### 2.3.4. Capillary Foam

Capillary foam is produced on the basis of Pickering foam, and the latter can be regarded as the precursor of the former. The color wet foam and dry foam which are difficult to be prepared by traditional methods can be made, which greatly reduces the difficulty of product processing [66]. By adding the immiscible second liquid into Pickering foam, the bulk solution will be converted from viscous fluid to elastic gel to form capillary foam (as shown in Figure 5b). Capillary foam is a new type of foam material, which uses the synergistic effect of particles and a small amount of immiscible secondary liquid to achieve stability [68,69,70]. The second liquid is usually a polymer with high viscosity coefficient, such as gelatin, agar, xanthan gum, etc., but it is wrong to simply think it Pickering foam with polymer added, because the addition of the second liquid will spread at the gas–liquid interface, playing the role of bridging particles and promoting the formation of a network. At the same time, the addition of the second liquid can also play an auxiliary wetting role, which makes the fibers more easily absorbed on the gas–liquid interface, maintain stability and increase the viscosity of the system. After the foam is dried, the fibers serve as the main skeleton of the foam, which mainly plays a supporting role. After the capillary foam added with the second liquid is dried, the fibers pile up more compactly, and the overall mechanical properties are improved, so that the foam can maintain its original network structure after drying. In 2014, Ahsan et al. [71] used microcrystalline cellulose as stable particles and added polymer chitosan, and found that the composite dispersion appeared gelation behavior at pH = 7. The synergistic effect of chitosan and MCC (Microcrystalline cellulose) enhanced molecular entanglement and produced synergistic network structure. As a result, foam with porous structure was prepared under this condition, and its energy absorption was as high as 32.86 kJ/m^3^.

### 2.4. Preparation of MFC-/NFC-Based Porous Solid Foam

The preparation of porous solid materials mainly includes three steps (Figure 6): the first step is sol-gel transformation (gelation), the second step is network perfection (aging), and the third step is gel transformation (drying stage). Firstly, the precursor material is dispersed in appropriate liquid to form colloids, and then bubbles are introduced to make wet foam. During the drying process, the foam begins to age and the foam network structure is gradually shaped. Cellulose-based porous foam material is obtained by removing solvent from wet foam, and its drying process determines the porosity, density and other characteristics of the final solid foam. It is a big problem to ensure that the foam can maintain its original porous network structure after drying, without deformation and collapse. When moisture evaporates, it will produce stress caused by capillary force, which will cause the deformation of voids to cause foam rupture, warping and collapse [72,73]. In view of this problem, many scholars have developed some drying technologies to minimize the damage degree of foam porous network structure, but each method has its own advantages and limitations. It can be seen from Table 2 for details. At present, the most commonly used methods are natural drying or oven drying, freeze-drying and supercritical drying, among which freeze-drying is the most rapidly developed, but freeze-drying is not suitable for large-scale production because of its high energy consumption and high cost. Supercritical drying has high requirements in terms of equipment and a complex operation, which is not suitable for industrial production. Natural drying or oven drying has simple operation and low equipment requirements, but the pore size distribution of the obtained foam is not uniform enough. Therefore, the drying method is one of the bottlenecks in the development of solid porous materials at present. Many people take different optimization methods to solve the disadvantages of the above processes in order to obtain high-porosity solid foam materials with good properties (Table 3) by a simple and low-energy drying method.

#### 2.4.1. Freezing Drying

Freeze-drying mainly uses the principle of sublimation to freeze wet foam at low temperature, and then sublimes it under vacuum. After sublimation of ice crystals, holes will be left, and porous solid foam materials can be prepared. The freezing speed plays an important role in the microstructure and compressibility of solid foam [4]. In 2017, Gupta et al. [74] focused on the use of ice-templates (after freezing of suspensions and sublimation of the formed ice crystals) to produce porous cellulose based materials, and discussed the influence of processing conditions such as temperature gradient and freezing rate on foam morphology during the freezing process In 2008, Pääkkö et al. [5] reported for the first time that aerogels with good electrical conductivity were prepared by two different freeze-drying methods: low temperature and vacuum. In low temperature freeze-drying, the mold containing hydrogel was quickly immersed in liquid propane at −180 °C, and then the frozen samples were placed in a vacuum oven until the pressure was kept at 1 × 10^−2^ mbar. In vacuum freeze-drying, the mold filled with hydrogel was put into a vacuum oven, the gel was frozen quickly, and the ice crystals were sublimated until the pressure was kept at 1 × 10^−2^ mbar. The aerogel prepared by the above two methods has a porosity of 98% and a density of 0.02 g/cm^3^, in which the specific surface area of low-temperature freeze-drying is 66 m^2^/g and that of vacuum freeze-drying is 20 m^2^/g. When the freezing speed is low, ice crystals will nucleate and grow spontaneously, while when the freezing speed is high and the temperature is low, the number of ice crystal nuclei is large, the volume of ice crystals is small, and the gaps formed are smaller and uniform. When freezing at low temperature, water pushes MFC and NFC in the process of forming ice crystals, and finally gathers between ice crystals. When ice crystals volatilize, dense porous structures will be formed, and MFC and NFC act as skeletons in the whole structure, thus ensuring the mechanical properties of solid foam. Subsequently, more and more people adopt similar methods to prepare aerogels with various properties [2,75,76]. Han et al. [47] put the fast freeze-drying flask filled with NFC and CNC (cellulose nanocrystals) suspensions with different concentrations in an ultra-low temperature freezer at −75 °C for two hours, and then transferred it to a freeze dryer for vacuum freeze-drying at −88 °C for three days, thus obtaining a layered foam structure with a pore size of 0.5–3 um. Compared with CNC, the foam fracture surface made by CNF with larger size is not as smooth as the former, and the cellulose concentration and fiber surface charge will affect the self-assembly behavior of fibers. Korhonen et al. [77] dried the MFC hydrogel in vacuum, and coated its surface with hydrophobic modification. Finally, the ultra-light selective oil absorption material with density of 20–30 mg/cm^3^ and porosity >98% was obtained.

In order to further reduce the entwined degree of fiber during freeze-drying and increase the specific surface area of the material, Sehaqui et al. [78] proposed to replace the traditional freeze-drying with tertiary butyl alcohol. The results showed that the capillary action was lower in the presence of tert-Butyl alcohol (TBA), and the surface area of aerogel obtained was as high as 332 m^2^/g. In the prior art, freeze-drying is a versatile method for preparing foam biomaterials with controllable structure, and the properties of the materials can be easily adjusted by controlling their microstructure, so it is widely used [3]. However, due to its high price, industrial production has not yet been realized. In order to reduce the problem of high energy consumption of the freeze drying Josset et al. [53] proposed a straightforward freeze—thawing—drying procedure; the method is based on urea is complementary fertilizer increased the rate of ice nucleation, the MFC/urea suspension using ice template steps under −45 °C, and then thaw dehydration in the environment, and finally drying at 105 °C. This method solves the problem of the high cost of freeze-drying, and urea will be dissolved in water during dehydration, which has the potential for reagents recovery and utilization. Foams and aerogels prepared by freeze-drying have nanometer and micron pore sizes and good porosity, which can be used in the fields of drug slow release, functional wall plywood, water-oil separation and so on.

#### 2.4.2. Supercritical Drying

In addition to freeze drying, supercritical drying is also a common method to realize high specific areas. It adopts a dry medium to replace the original solvent into supercritical fluid under critical temperature and pressure conditions, and then releases the fluid slowly under reduced pressure. Because supercritical fluid is a fluid between gas and liquid, the original gas/liquid interface no longer exists, and there is no pressure attached to capillary force, thus avoiding the deformation and collapse of foam structure caused by capillary force [79]. Commonly used drying media are methanol, ethanol and carbon dioxide. Because methanol and ethanol are flammable and explosive, carbon dioxide is widely used as drying media on a large scale. Compared with freeze-drying, the solid foam prepared by supercritical drying avoids the problem of aggregation among freeze-drying fibers, and its density is relatively low, the surface area is increased, and the pore size is mainly distributed in nanometer level (Table 3). Compared with freeze-drying, supercritical drying is not so widely used. The specific surface area value distribution range of the prepared porous materials is limited, mainly between 200–300 m^2^/g, which is different from freeze-drying (specific surface area value difference is relatively large). Deniz et al. [79] adopted a multi-stage solvent exchange process, in which ethanol is used instead of water, and then ethanol is removed by supercritical CO_2_ to obtain aerogel with a density of 0.009 g/cm^3^. Freeze-dried aerogels (density: 0.023 g/cm^3^) were also prepared from the same materials. It was found that light white spongy aerogels were obtained after removing water, and the structure did not collapse obviously. The aerogels dried by SCCO_2_ (supercritical carbon dioxide) had lower density and higher specific surface area. Wu et al. [6] used a supercritical drying method to dry cross-linked foams and aerogels. The specific surface area was as high as 430 m^2^/g, and the Young’s modulus was as high as 711 kPa through physical and chemical cross-linking, which showed good mechanical properties. At the same time, the layered double void structure of the material was obtained through adsorption experiments on silver ions, which gave it considerable adsorption capacity and enabled it to be applied in antibacterial fields. Aerogel prepared by supercritical drying can maintain a good network structure because there is no gas-liquid interface in the drying process, and its pore size is mostly nanoscale, but the requirements for equipment are relatively high. At the same time, toxic gases may be produced during solvent replacement, therefore, it is necessary to control the speed of fluid release during operation, and its development is limited due to high cost and complex operation.

#### 2.4.3. Oven Drying

Compared with the above two methods, oven drying is the simplest and the cost is lower, which is suitable for industrial production. However, the main problem at present is that the foam structure collapses seriously during the drying process, and the optimization of drying process is not perfect at present. Most of them are prepared by improving the stability of wet foam (such as Pickering foam and capillary foam), and then combined with oven drying. The aperture is mostly micron- or even millimeter-scale, otherwise it is inhomogeneous and large cavities can easily appear in the material. To solve this problem, Cervin et al. [28] used sulfite softwood dissolving pulp as raw material, prepared NFC by TEMPO oxidation combined with high-pressure homogenization, and then added octylamine to make NFC better adsorbed on the gas-liquid interface. After mechanical foaming, the wet foam was placed in porous ceramic frit (soaked overnight, water filled the whole gap), and then dried in an oven at 60 °C. The density of the foam was 13 kg/m^3^, and the porosity was as high as 99%. In the drying process, the ceramic plate is equivalent to a water reservoir, which can provide liquid flow for the foam, avoid the intercommunication of liquid in the cavity and the rupture area, and reduce the generation of independent air pockets, thereby improving the non-uniformity of the pore diameter of the solid foam. In other literature by the author, it is shown that the addition of octylamine makes the fiber more adsorbed on the gas-liquid interface, and there are more counter ions around it, and the disjoining pressure between bubbles increases, which prevents bubbles from coalescence and disproportionation [7]. Liu et al. [8] prepared MFC as stable particles from bagasse, and formed capillary foam by adding secondary liquid (gelatin) into bulk solution, and dried it in an oven (40 °C–60 °C) overnight, and finally prepared porous fiber foam with light weight and high porosity.

## 3. Properties of Porous Foam Materials

### 3.1. Surface Area and Porosity

Porous foam materials can be used in drug carriers, adsorption and electrodes because of their light weight, low density and high specific surface area [2]. In order to improve the high specific surface area (SSA) of materials, it is very important for the specific surface area of porous materials that NFC is well dispersed in solvent mixture. In order to prevent the fibers from gathering together to form larger fiber bundles, which will lead to the increase of material density and decrease of SSA [82], the density of materials can be reduced and the specific surface area can be increased by adding octylamine or changing its concentration [28]. The most significant factors affecting the properties (density, specific surface area and porosity, etc.) of porous materials are the initial fiber concentration and drying process. Nissila et al. [83] confirmed that the initial concentration of NFC is directly proportional to the density of aerogel and inversely proportional to porosity and specific surface area. Sehaqui et al. [78], after mixing, centrifuging and removing supernatant before freeze-drying, firstly exchanged solvent between water and ethanol, then exchanged solvent between ethanol and tert-Butyl alcohol (TBA), and found that the specific surface area of aerogel finally made from fiber was 332 m^2^/g. At the same time, it has been suggested that supercritical CO_2_ drying will produce higher specific surface area than freeze drying. Deniz et al. [79] adopted two methods to prepare aerogels at the same time, and the results obtained confirmed this view. The specific surface area of supercritical CO_2_ drying was 115 m^2^/g, and the porosity was as high as 99%, while that of freeze drying was 20 m^2^/g, and the porosity was 96%. The BET (Brunauer–Emmett–Teller) method and BJH (Barrett–Joyner–Halenda) method are mainly used to calculate the specific surface area and pore size. In the freeze-drying process, due to the shrinkage and collapse of small pores, the BJH method (pore size 1.7–300 nm, Heath and Thielemans, 2010) is more suitable [79].

### 3.2. Mechanical Properties

The mechanical properties of porous foam materials are mainly manifested in hardness, compressive strength, etc. Secondly, in some application fields, the ability to resist deformation is also very important, and the properties of materials often have a great relationship with the shape of materials. Some studies have reported that there are three stages in the compression deformation of porous solids [2,38,72]: (a) under yield stress, the elastic deformation of the material is linear, which is mainly caused by the elastic deformation of cell wall bending and the compression of large pores; (b) the plateau area changes from elastic deformation to plastic deformation. At this stage, with the increase of material deformation, the stress no longer changes, which means that the cell begins to collapse, and the structure suffers damage and no longer has resilience; (c) densification region, the cell walls are in contact with each other, and the porous structure is seriously collapsed. Based on this feature, Chen et al. [38] prepared an anisotropic whole biological CMC (carboxymethyl cellulose)/NFC aerogels by directional freeze-drying, which showed honeycomb structure. It was confirmed by compression and three-point bending tests that there were linear elastic regions with 30% elastic strain and 30–70% strain was plateau regions. Then came the densification region (70–90%), and the compression modulus in the vertical direction (8.7 MPa) was significantly higher than that in the horizontal direction (1.5 MPa). In addition, Kobayashi et al. [84] reported a new type of three-dimensional ordered NFC aerogel with a surface area of 500–600 m^2^/g. The material showed a linear elastic region under 10% strain, and began to deform plastically after reaching the yield point. When the strain reached 60%, the densification region appeared and the skeleton became dense. Compared with ordinary aerogels, it had good optical properties and mechanical toughness, and its elastic modulus increased linearly with the increase of density, reaching a maximum of 1MPa. But most of the cellulose-based foams do not exhibit a compression behavior similar to that of the platform (as shown in Figure 7). This is related to the use of various processing routes. Cervin et al. [7] have created lightweight and strong NFC foam using Pickering foam processing route, they conducted compression experiments of NFC foam with a density of 0.05 g cm^−3^ and a porosity of 96.7%. Research has shown that that there was no platform area in this experiment. At 60% strain, the linear viscoelastic region appeared, and then followed densification region directly. The Young’s modulus is 437 ± 63 kPa according to the slope under low strain, and the compressive energy absorption is 48 ± 11 kJ m^−3^ at 80% strain. As a result, we can see that the mechanical properties of the solid foams produced by this process are comparable to other types of NFC foams, but the strength is lower than that of ordinary plastic foams, such as EPS (expanded polystyrene board) whose Young’s modulus is as high as 6000 kPa. The report also showed that the mechanical properties of the foam can be adjusted by changing the chemical composition and foaming composition of the NFC. A similar problem arose in Sehaqui’s [2] report, which found that the platform area where a typical elastoplastic polymer foam should be found did not appear in the report. In this paper, the porous MFC-based foams with high mechanical strength were prepared by freeze drying without solvent. Although the transition from linear stress-strain behavior to non-linear stress-strain behavior was gradual, it is clear that materials exhibit a collapse behavior that often results from the formation of plastic hinges due to plastic yielding of the cell wall materialization. At higher strains, the porous structure of the foam becomes densification, and the cell walls contact each other which result material to rigid. Wu et al. [6] have prepared the dual-porous NFC aerogels with good mechanical properties and high stability in water by physical/chemical cross-linking, the hydrophilicity and mechanical strength of aerogels can be precisely controlled, the most important of which is that the aerogels have dual-porosity. In comparison with other cellulose porous materials, the properties of macro-porosity and high strength of NFC foams are combined with the properties of Meso-porosity and high surface area of NFC aerogels. Both horizontally and vertically, the mechanical strength of the cross-linked materials is higher, and the mechanical strength of the cross-linked materials increases with the content of the NFC. Generally speaking, the mechanical strength of porous materials is inversely proportional to the network pore size, and the pore size mainly depends on MFC/NFC concentration and drying technology. It can be seen from Table 3 that compared with the other two drying methods, supercritical drying can form small and uniform pores. Lee et al. [85] prepared a unique and controllable microfiber porous foam by a unidirectional freezing method. When the microfiber concentration increased from 2% to 8%, the compressive stress was greatly increased (from 30.7 kPa to 366 kPa).It was found that the foam morphology changed with the increase of microfiber concentration. At low concentration, the foam showed a cross-linked network structure. With the increase of concentration, the cross-linked network gradually changed to a layered channel structure. When the concentration was 8%, the foam had formed a highly ordered channel structure parallel to the freezing direction, and the cell wall thickness increased.

## 4. Applications

### 4.1. Sound Insulation and Thermal

MFC/NFC acts as a skeleton in the whole structure, which endows foam with certain mechanical ability. Meanwhile, because fibers overlap each other to form a staggered pore structure, this unique structure makes cellulose-based porous foam material occupy unique advantages in heat insulation and sound insulation. Based on the above characteristics, it has broad application prospects in the field of wall construction. For this porous material, most of the volume is air, and the heat transfer mechanism mainly depends on gas phase conduction. The smaller the pore size, the more restricted the movement of gas molecules, so the slower the heat transfer. Wang et al. [86] prepared ultra-light composite foam with nanocrystalline cellulose (NCC), polyvinyl alcohol and the cross-linking agent BTCA (1,2,3,4-bButanetetracarboxylic acid), and the stress was 18 times that of pure NCC foam under the same conditions. The thermal conductivity was 0.027 Wm^−1^ K^−1^. Scanning electron microscopy (SEM) shows that the foam has a highly disordered lamellar assembly structure with a pore size of 50–100 um. The regular cell wall and pores in the composite foam improve the thermal insulation and mechanical properties of the composite foam. In addition to the inherent structure of the material, the precursor materials (additives, crosslinking agents, etc.) also have great influence on the sound insulation and heat resistance of porous materials. Guo et al. [80] prepared the spongy NFC/BTCA/MDPA (2,2-dimethylol propionic acid) aerogel by cross-linking with the environment-friendly freeze-drying method. Based on traditional NFC aerogels, the aerogels were combined with MDPA and BTCA and other co-additives, which made them have good thermal insulation and flame retardant properties while incorporating the self-extinguishing properties, and the thermal conductivity was as low as 0.03258 Wm^−1^ K^−1^. The material presents a three-dimensional network structure with laminated layers, with NFC distributed among the layers. Its good thermal stability comes from the decomposition of MDPA during pyrolysis and the generation of acidic substances (phosphoric acid, etc.). These acidic substances further catalyze cellulose dehydration and form carbon structures at a lower temperature, thus preventing the formation of volatile substances. The material has a good development prospect in the field of thermal protection.

Porous materials can also be used as sound-absorbing materials in vehicles, buildings, household appliances, etc., which are mainly related to porosity, airflow resistance and tortuosity of materials. Porosity is the ratio of sample air volume to total sample volume, air flow resistance refers to the resistance to air as it enters the opening cell, and tortuosity refers to the shape of the channel in the material. As sound waves enter through the pores, the surface of the material vibrates with the air, and part of the energy is converted into thermal energy [87]. At the same time, scattering will occur on the fiber surface to absorb most of the energy, and the absorption of sound by closed cells is lower than that by open pores. Pöhler et al. [37] made a kind of porous sound-absorbing material with MFC and starch as raw materials. Its advantage is that compared with commercial sound-absorbing materials, it can achieve the same sound-absorbing effect with the least amount of material. At the same time, due to the addition of starch and other fillers, its mechanical properties have also been improved, and it can become a substitute for indoor sound-absorbing materials.

### 4.2. Medical Application

Cellulose-based porous foam has biocompatibility and can be regenerated and degraded, so it can be applied to biomedicine. At present, NFC aerogel has been tested in wound dressing, tissue engineering [88] and drug sustained release [89]. Haimer et al. [81] used bacterial cellulose aerogel as controlled release matrix loaded with bioactive substances (dexpanthenol and ascorbic acid), in which the loading amount depended on the concentration of loaded substances, and the release of active substances can be controlled by adjusting the thickness of aerogel according to the release curve. Most hemostatic dressings are sponge products, which have the ability to absorb blood, so as to trigger the exogenous coagulation mechanism, in which porosity and water absorption are important indicators. Lu et al. [14] separated NFC from wood powder and oxidized it into dialdehyde NFC by sodium periodate. The composite aerogel was prepared by the chemical crosslinking of dialdehyde NFC and collagen. The porosity was 90–95% and the water absorption rate was as high as 4000%, which provided enough liquid holding space for wound dressing. Cai et al. [16] prepared NFC ultra-light porous aerogel microspheres by liquid nitrogen quick freezing and freeze drying. Its unique structure gives it good water absorption ability. At the same time, cell culture experiments confirmed that this material can promote cell differentiation and proliferation as well as transfer of metabolic waste, and has potential to become a new material in tissue engineering. On this basis, Liu et al. [17] prepared aerogels with three-dimensional structures, whose swelling degree can reach 500 times, porosity up to 99.7%, specific surface area of 308 m^2^/g, which is an ideal tissue engineering material and can be used in biological scaffolds. Moreover, through simulation experiments, it has been proved that this material can promote the survival and proliferation of tumor cells.

### 4.3. Packaging Materials

The MFC/NFC-based porous foam material has a good honeycomb structure, which can form a cell network with the least amount of material per unit volume, and is like an arch bridge on the whole. Therefore, porous materials are endowed with good mechanical properties. On the one hand, the foam material is based on cellulose, and the raw material is rich and biodegradable. On the other hand, its unique network structure can absorb certain energy because of the existence of a large number of holes, and has excellent buffering and vibration isolation function, based on these two points, some scholars have developed a strong interest in cellulose-based cellular foam for shock absorbers in packaging. Using freeze-drying, Svagan et al. [90] prepared cellulose and starch bio-polymerized foam, because MFC form a strong network structure in the matrix material, the foams exhibited a Young’s modulus of 6.2 GPa and a tensile strength of 160 MPa. The results show that it is a potential substitute for petroleum-based polymers because of its absorption of impact energy in packaging materials and its good cushioning effect in sandwich structure core solid materials and car seats. Sinthu et al. [91] prepared cellulose aerogel via the sodium hydroxide solution system, and then filled the nano/micron pores of the aerogels with a reactive index matching PMMA (polymethylmethacrylate) to obtain cellulose tablets with high transparency and high elongation at break, and its expansion coefficient is 10.45 ppmK^−1^, which is obviously lower than that of plastic (79.70 ppmK^−1^ polymethyl methacrylate). The material can be used in bouquet packaging due to its good transparent appearance and mechanical properties. Similarly, a cellulose aerogel with high flexibility and transparency was prepared by Qin et al. [92]. With growing awareness of environmental protection, the requirements of packaging ensure they must not only have good mechanical properties, but also be degradable, recyclable and have other characteristics, and so cellulose-based foam has a good application prospect in packaging.

## 5. Conclusions and Outlook

Cellulose-based porous foams are light and strong, multi-porous materials and different properties can be obtained by different preparation methods. Foam is a complex multi-phase coexisting system, so it is very important to maintain its stability. At present, research on the stability of wet foam is not sufficiently in-depth. In previous studies, NFC and MFC played the role of stabilizing particles in the whole system, but how this affects the dynamics of the gas–liquid interface is not understood comprehensively. At the same time, how to ensure the minimum damage degree of the original network structure of foam is still a huge challenge facing the whole industry. Although some drying technologies (freeze-drying, supercritical drying) have been developed to ameliorate this problem, the scale expansion is limited by cost and practical feasibility, which maintains this kind of material still in the research and development stage and it has not been put into actual production. Simplifying solvent exchange and the supercritical drying process can be the focus of future research, and the related changes of internal structure and stress of materials during drying can also be further explored. On the other hand, research on cellulose-based porous foam materials is mostly limited to biomedicine, liquid adsorption, thermal insulation, etc., and research on micro-electronic products such as electrical equipment is still limited, but is expected to develop. The selection of foam/aerogel precursor materials is very important for endowing materials with new characteristics. We hope to develop more new materials and process routes in the future to enrich and improve various lightweight materials and finally apply them in practice.

## Figures and Tables

**Figure 1 materials-13-05568-f001:**
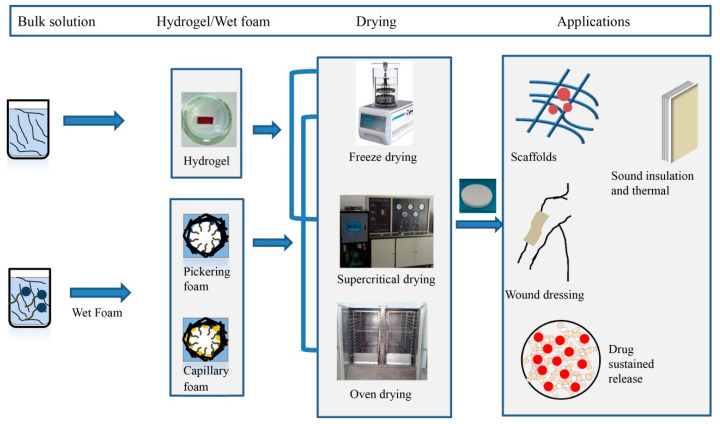
The stability approach and main application of cellulose-based porous materials.

**Figure 2 materials-13-05568-f002:**
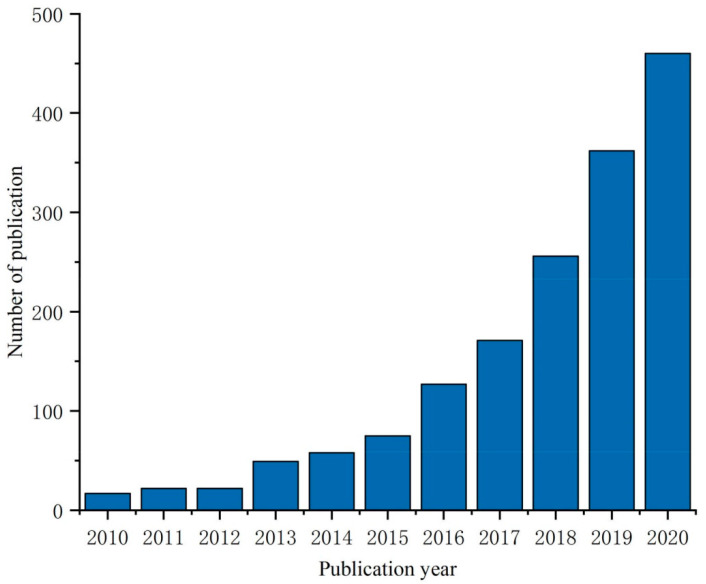
The annual amount of scientific publications obtained through Science Direct with the subject of “cellulose foam; cellulose aerogel”.

**Figure 3 materials-13-05568-f003:**
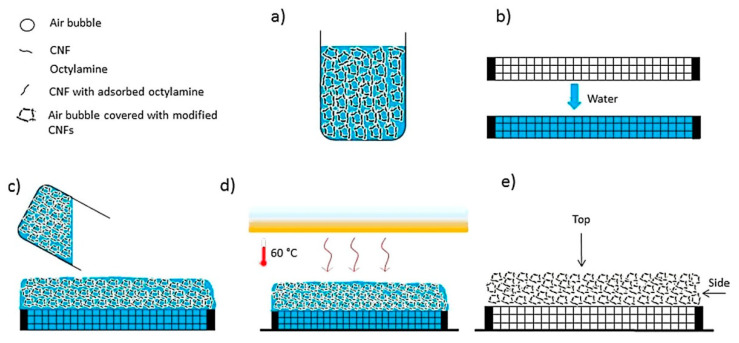
Schematic illustration of the process of the nanofibrillated cellulose (NFC)-stabilized solid foams: (**a**) NFC suspension with a content of 1 wt%, (**b**) dry porous ceramic frit, (**c**) wet foam on top of ceramic frit, (**d**) oven drying at 60 °C, (**e**) solid foam after drying. (Adapted from Cervin and Johansson [28] with permission from Acs Appl Mater Interfaces).

**Figure 4 materials-13-05568-f004:**
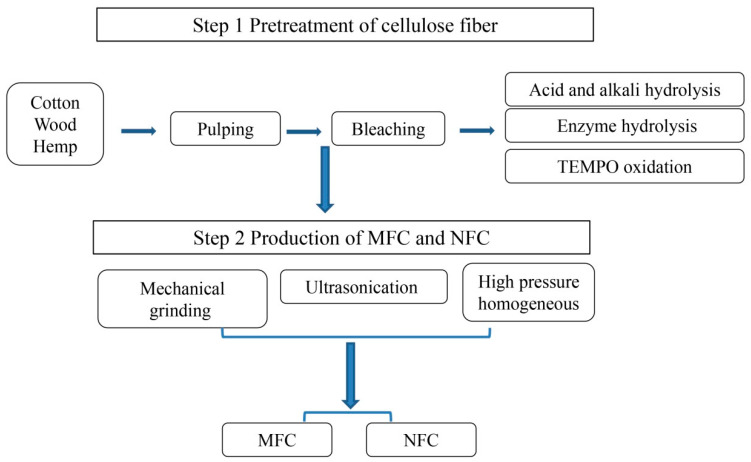
The steps of extracting microfibrillated cellulose (MFC) and NFC from natural plants.

**Figure 5 materials-13-05568-f005:**
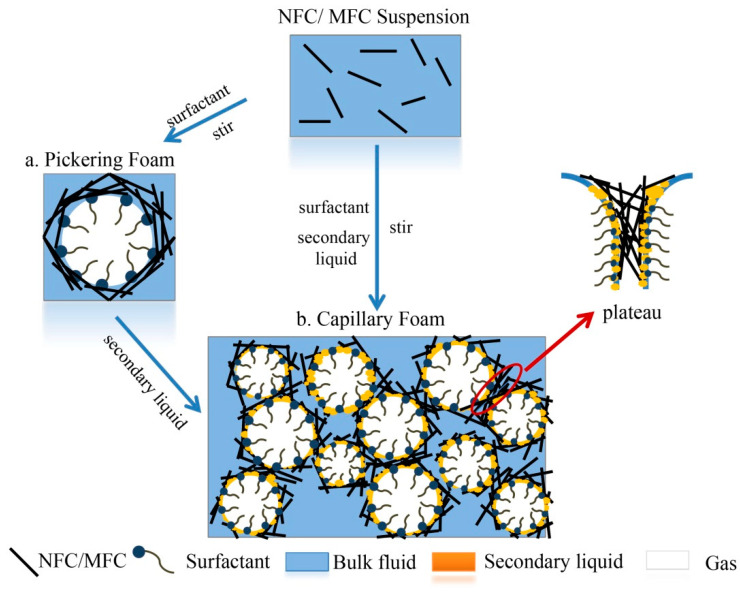
The stabilization mechanism of wet foam: (**a**) the structure of Pickering foam; (**b**) the structure of capillary foam.

**Figure 6 materials-13-05568-f006:**
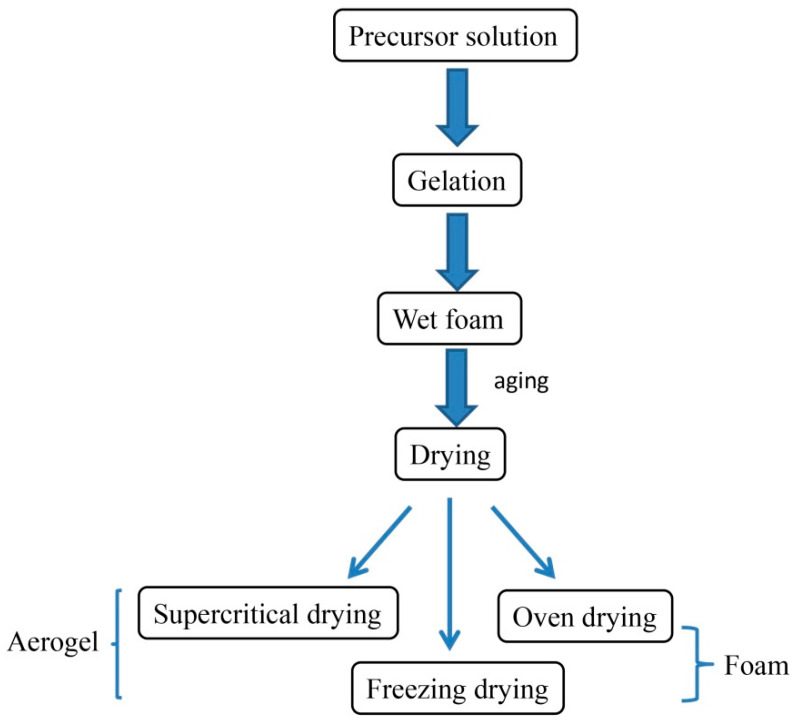
A flow chart on preparation of porous solid materials.

**Figure 7 materials-13-05568-f007:**
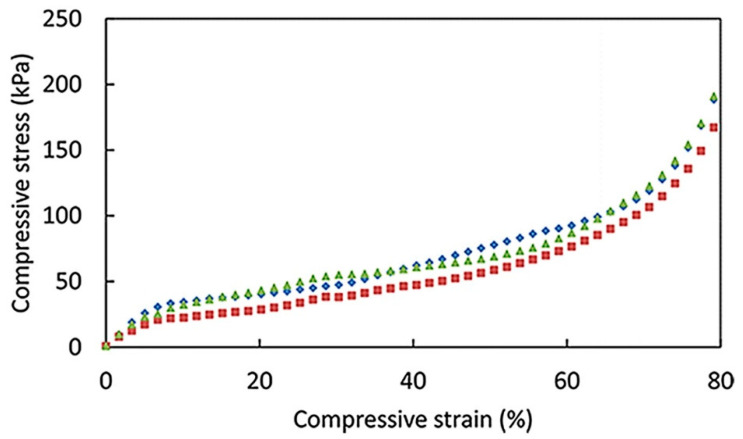
Stress−strain curves in compression for NFC (1 wt%) stabilized foam. The three samples are taken from different positions in the same dry foam (Adapted from Cervin and Johansson [7] with permission from Acs Appl Mater Interfaces).

**Table 1 materials-13-05568-t001:** Comparison of different preparation methods with different raw materials.

Materials	Name	Method	Particle	Reference
Kraft pulp bleached from coniferous wood	NFC	TEMPO Mechanical	D: 3.0 ± 0.3 nm L: 1.2 ± 0.4 nm	[43]
Bleached Kraft pulp	NFC	—	D: 35–40 nm L: 2.3 mm	[44]
—	NFC	TEMPO	D: 4 ± 1.4 nm L: 255 ± 104 nm	[45]
Kraft pulp bleached from coniferous wood	MFC	Mechanical	D: 30 um L: 2.3 mm	[37]
Bleached sulfite pulp from spruce	NFC	Mechanical	D: 30 um L: 2–3 mm	[46]
Commercial eucalyptus bleach pulp	NFC	TEMPO Homogeneous	D: 20–50 nm	[3]
Birch bleached Kraft pulp	NFC	Enzyme Homogeneous	D: 20 um L: 150–300 um	[3]
Bleached sulfite softwood cellulose	NFC	Enzyme Homogeneous	D: 5–10 nm, 100–500 nm	[5]
Bleached wood pulp	NFC	Acid Homogeneous	D: 16 ± 4 nm, 21 ± 7 nm L: 616 ± 200, 732 ± 208 nm	[47]
Bleached Kraft pulp	MFC	Fenton Mechanical	D: 10–100 um L: 0.2–7.5 mm	[48]
Bleached hardwood Kraft pulp and soft acid bagasse sulfite pulp	MFC	Enzyme Mechanical	D: 60 um	[49]

D: diameter of cellulose; L: length of cellulose.

**Table 2 materials-13-05568-t002:** A comparison of drying approaches for various solid foam.

Name	Method	Particle	Advantages	Disadvantages
Supercritical drying	Replacing the solvent with supercritical fluid (methanol, ethanol and CO_2_)	Nanosize	Dimensions stay in nanosize	Expensive and complicated method
Freezing drying	Precooling at −4 °C and freezing in liquid nitrogen then freezing overnight in a frozen drying oven	nm to um	Establish a good network structure	Expensive
Oven drying	Drying in the oven at 105 °C for 24 h	um to mm	Well established for the industry	Severe structural collapse

**Table 3 materials-13-05568-t003:** Comparing the properties of porous materials prepared by different drying methods.

Name	Method	Parameter	Advantage	Application Reference	
Aerogel	Freeze drying	514.15 m^2^/g 2–10 nm	Good circulation stability	High performance super capacitors	[43]
Foam	Freeze drying	35.8 m^2^/g	Controllable structural	Thermal insulation	[9]
Aerogel	Liquid nitrogen and freeze drying	40.31 m^2^/g 13.66 nm	Thermal resistance and high tenacity	Personal protectable equipment	[80]
Foam	Oven drying	300–500 μm 98%	Lightweight and strong	_	[7]
Aerogel	Supercritical carbon dioxide drying	200 m^2^/g	_	Drug sustained release	[81]
Aerogel	Supercritical drying	130–160 m^2^/g 0.5 ± 0.2 μm	Good hygroscopic property	wound dressing	[44]
Aerogel	Freeze drying	20–30 m^2^/g 24.7 ± 10.4 μm	_	_	[44]
Aerogel	Supercritical carbon dioxide drying	240–280 m^2^/g 0.86–0.92 μm 91–96%	Sound absorption High intensity	Office ceiling	[13]
Foam	Freeze drying	93–99.5%	Porous and excellent mechanical properties	New composite material for energy absorption	[2]
Aerogel	Oven drying	70–120 m^2^/g	Excellent mechanical properties	Green heat insulation building materials water purification material	[12]
Aerogel	Freeze drying	70 m^2^/g 98%	High strength and deformation	Functional conductive material	[5]
Foam	Oven drying	99.6%	Water resistance and wet elasticity	Water and oil separation	[28]
Aerogel	Liquid nitrogen and freeze drying	99.38–99.97%	Controllable structural	_	[48]
Aerogel	Liquid nitrogen and freeze drying	11–42 m^2^/g 99.8–99.1%	Super hydrophobic	Water and oil separation	[29]

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
