# Peer review of "MFC/NFC-Based Foam/Aerogel for Production of Porous Materials: Preparation, Properties and Applications"

_materials, 2020, doi:10.3390/ma13235568_

Round 1
Reviewer 1 Report
In the paper is described an overview of cellulose foam/aerogel obtaining as method for manufacturing of porous materials with different applications. The foam forming is a novel technique for producing bulk materials with low density and high porosity. It is a comprehensive approach well structured.
However, I have some comments and suggestions:
- In the field of cellulose fibres foams studies were developed early 1960s and intensive studied in the last years due to the potential savings in terms of raw materials, energy and water comparing with conventional water-forming fibrous materials. Moreover, these materials are environmentally friendly alternatives to existing petroleum based foams. Based on the fact that the cellulose foams can be obtained with different type of cellulose fibres (included secondary cellulose fibres) I consider that a subsection with description of obtaining cellulose fibre foams must be included in the manuscript. Generally, NFC and MFC are used especially to improve the strength properties of cellulose fibre foams materials (being expensive materials).
- In this context the section 4 Application, will be completed with details about application of cellulose fibres foams materials in sound and thermal insulation (not only NFC and MFC foams).
- The authors do not mention anywhere in the manuscript about the application of cellulose fibres foams/aerogel in the Packaging as shock absorbers. I suggest to complete this section with some details regarding the application of these materials in packaging.

Reviewer 2 Report
In this review manuscript the recent research progress and the potential application prospects of porous cellulose foam materials are discussed. In my opinion, this manuscript needs to be thoroughly revised before it can be approved
The title of this manuscript has to be rewritten. Cellulose-based foams are a broad area and using this term suggests that the subject would be discussed more broadly in the text. Now the focus of this manuscript is only in NFC/MFC foams, which should be mentioned in the title.
The authors have shown in Figure 2 number of publications with the subject of “cellulose foam; cellulose aerogel”. According to this data, lot of publications related this topic have been published in years 2019 and 2020. However, the authors have only cited ~15 publications of these years. A clearly larger number of the recent publications should be cited. The number of citations (73) is also quite low considering that this is a review article. In addition, the references need to be checked and mistakes need to be corrected e.g. “Schmitt” (line 174), “Liu” (line 221).
Section 2.2 Discusses more about foaming agents. What other inert gases can be used in addition to nitrogen? Please provide also examples of chemical foaming agents. How the foaming methods affects to the foam quality e.g. stability/density/porosity?
Section 3 “Properties of porous materials” and section 4 “Applications” are very short. These topics should be much more comprehensive, as these are, however, the main themes of the publication
Minors corrections:
- Explain abbreviations NFC/MFC (line 27), EVA/EPS (line 56), SDS (line 185) and all others, when mentioned first time
- Line 72: specify Science Direct database
- Texts in Figure 4 are difficult to read and should be rotated
Reviewer 3 Report
Lignocellulosic raw materials have aroused great interest in scientists and packaging manufacturers in recent years. Therefore, the subject matter presented in the article may be of particular interest to the readers. The authors presented in the article the possibilities of using nanofibrillated cellulose and microfibrillated cellulose in the production of aerogels and foams and the problems encountered in the production of these materials.
Reviewer 4 Report
This review article deals with the fabrication, properties and applications of nanocellulose foams and aerogels. Here are my main remarks at this stage of the reviewing process:
- In the introduction part, the authors should better explain and justify the main interest of their paper with regards to the others review articles already published on this topic. For instance, the review article by Gupta et al. Appl. Sci. 2018 published in MDPI is not mentioned/discussed in this study. I think that this study could be complementary since it provides additional information, in particular regarding the fabrication and properties of “pickering” and “capillary” nanocellulose foams.
- 1 is not clear and inaccurate. It is possible to produce nanocellulose foams without adding surfactant in the suspension. The pictures showing potential applications of nanocellulose foams are not clear. I do not understand the drawings. Please add photographs of real materials.
- 3 is inaccurate. From this figure, we understand that NFC are produced from MFC. This is not always the case. We can produce NFC from cellulose fibres using chemical treatment (e.g., TEMPO mediated oxidation) combined with mechanical grinding (or sonication). The difference between MFC and NFC is mainly related to the size of extracted cellulose nanofibers. Please improve Fig. 3 and the related comments.
- The authors should add relevant figures and comments regarding the porous microstructure of nanocellulose foams and in particular those of pickering and capillary foams. What are the main features of pickering foams in comparison with nanocellulose foams obtained following others processing routes?
- 6 is not relevant at all. Indeed, most of nanocellulose foams does not exhibit a plateau-like compression behaviour (as many classical foams) but a strain hardening behaviour. Instead of using a schematic diagram, the authors should provide “real” compression curves of nanocellulose foams obtained using various processing routes (in particular nanocellulose foams obtained using the processing route shown in Fig. 4) and discuss the links between the processing routes, the microstructures and the mechanical compression behaviour.
- In part 4 of the manuscript, the authors should discuss the interest of using capillary and pickering foams (keeping in mind the peculiar porous microstructure and mechanical and physical properties of this type of nanocellulose foams. It would be of great interest for the reader.
Round 2
Reviewer 2 Report
I think that the manuscript has now been corrected according to the suggestions. I would recommend the acceptance.
Reviewer 4 Report
The authors have made significant changes in order to improve the manuscript content. Authors must carefully check the text of the manuscript to improve english language and style. In addition, several parts in the manuscript are still rather difficult to understand or unclear.